# A Potent Inhibitor of the Cystic Fibrosis Transmembrane Conductance Regulator Blocks Disease and Morbidity Due to Toxigenic *Vibrio cholerae*

**DOI:** 10.3390/toxins14030225

**Published:** 2022-03-18

**Authors:** Fabian Rivera-Chávez, Bradley T. Meader, Sinan Akosman, Vuk Koprivica, John J. Mekalanos

**Affiliations:** 1Division of Host-Microbe Systems and Therapeutics, Department of Pediatrics, University of California, San Diego, CA 92093, USA; fcrivera@health.ucsd.edu; 2Division of Biological Sciences, University of California, San Diego, CA 92093, USA; 3Department of Microbiology, Harvard Medical School, Boston, MA 02115, USA; bradley_meader@hms.harvard.edu; 4Vanda Pharmaceuticals, Inc., Washington, DC 20037, USA; sinan.akosman@vandapharma.com (S.A.); vuk.koprivica@vandapharma.com (V.K.)

**Keywords:** cholera toxin, *Vibrio cholerae*, diarrheal syndromes, anti-infective drug therapy

## Abstract

*Vibrio cholerae* uses cholera toxin (CT) to cause cholera, a severe diarrheal disease in humans that can lead to death within hours of the onset of symptoms. The catalytic activity of CT in target epithelial cells increases cellular levels of 3′,5′-cyclic AMP (cAMP), leading to the activation of the cystic fibrosis transmembrane conductance regulator (CFTR), an apical ion channel that transports chloride out of epithelial cells, resulting in an electrolyte imbalance in the intestinal lumen and massive water loss. Here we report that when administered perorally, benzopyrimido-pyrrolo-oxazinedione, (R)-BPO-27), a potent small molecule inhibitor of CFTR, blocked disease symptoms in a mouse model for acute diarrhea caused by toxigenic *V. cholerae*. We show that both (R)-BPO-27 and its racemic mixture, (R/S)-BPO-27, are able to protect mice from CT-dependent diarrheal disease and death. Furthermore, we show that, consistent with the ability of the compound to block the secretory diarrhea induced by CT, BPO-27 has a measurable effect on suppressing the gut replication and survival of *V. cholerae*, including a 2010 isolate from Haiti that is representative of the most predominant ‘variant strains’ that are causing epidemic and pandemic cholera worldwide. Our results suggest that BPO-27 should advance to human Phase I studies that could further address its safety and efficacy as therapeutic or preventative drug intervention for diarrheal syndromes, including cholera, that are mediated by CFTR channel activation.

## 1. Introduction

As a global health burden particularly in developing countries, diarrheal disease has been estimated to account for nearly a million deaths a year and approximately 10% of all deaths in children under 5 years of age [1]. Cholera is the paradigm for an acute bacterial diarrheal disease and, unfortunately, the currently available vaccines for preventing this disease show little or no efficacy for young children where disease burden is most pronounced [2]. Cholera is a severe dehydrating syndrome that can kill victims within hours of the first onset of symptoms. *Vibrio cholerae*, the causative agent of cholera, has been well-studied in victims, animal models, and human volunteer challenge studies [3]. The disease itself can be entirely reproduced in experimental animals and in human subjects through ingestion of the bacterial protein called cholera toxin (CT). For example, ingestion of a few micrograms of CT in bicarbonate buffer can cause over 20 L of watery diarrhea in a healthy adult human subject, and this level of disease would otherwise be lethal without supportive care to replace fluids and electrolytes [1]. Thus, *V. cholerae* can be thought of as a living vector for delivery of CT. The fulminant diarrhea caused by the toxin has been demonstrated to enhance bacterial replication in the human host [4,5] and is thought to drive pathogen dissemination to secondary victims through fecal contamination of water and other fomites. By analogy, other bacterial pathogens such as enterotoxigenic *Escherichia coli* (ETEC) likely utilize CT-like toxins and heat-stable enterotoxins (STs) to drive their replication and dissemination within and between hosts [4,6]. While not as lethal, these enteropathogens remarkably cause diarrhea though mechanisms that are identical or analogous to CT through induction of luminal chloride secretion.

Pathogenic isolates of *V. cholerae* produce and secrete CT during the gut infection cycle. In natural settings, contaminated food or water transmits the organism by the fecal–oral route and if a high enough number of *V. cholerae* cells are ingested, some bacteria survive the acid barrier of the stomach and enter the upper intestine. Once in the small intestine, *V. cholerae* adhere to the intestinal epithelium, and this is where explosive replication of the organism then occurs. The toxin-co-regulated pilus (TCP) mediates these critical early bacterial colonization steps by mediating adherence to the intestinal mucosa [5], and remarkably, also serves as the receptor for infection of a filamentous bacteriophage CTXφ that encodes the CT subunits (CtxA and CtxB) [7]. Pathogenic strains of *V. cholerae* coordinately regulate production of the TCP intestinal colonization factor with CT in vivo. The contributory evolutionary processes that ultimately led to the emergence of toxigenic *V. cholerae* from apathogenic environmental strains through gene acquisition are not yet elucidated and continue to be a subject of investigation ever since the *V. cholerae* N16961 genome was sequenced and annotated in 2000 [8]. Epidemic strains are recognized as either classical biotype, those largely responsible in the first six pandemics from 1817–1923, or El Tor biotype, those dominant in the ongoing seventh pandemic that began in 1961 [9]. Phylogeny and historical records predict that both classical and El Tor strains first emerged separately from ancestral strains in Southeast Asia and then spread globally [1,9,10]. Furthermore, there are identified “waves” of closely related strains of *V. cholerae* with distinctive genomic differences and one group of these ‘variant’ strains has emerged to dominate the epidemiology of cholera globally [10]. For example, a single such variant strain caused the massive epidemic of cholera in Haiti in 2010 [11] and together with Yemen, these two countries alone have accounted for millions of cholera cases and tens of thousands of deaths [12]. Although there are effective cholera treatments such as fluid replacement therapy [13], these interventions can be difficult to implement under epidemic circumstances in the developing world. Thus, a simple orally administered drug intervention for blocking the lethal effects of cholera toxin (and related enterotoxins) is clearly an unmet medical need in the challenge to combat lethal bacterial diarrheal disease globally.

The molecular mechanisms for CT-induced diarrheal disease are well understood [14]. Production of CT by *V. cholerae* within the gut activates eukaryotic adenylate cyclase (AC) in the brush border of intestinal epithelial cells. Mechanistically, the CtxB subunits bind to their ganglioside receptors and mediate the translocation of the CtxA subunit into the eukaryotic cell cytosol where it then catalyzes the ADP-ribosylation of a regulatory subunit (a G protein called Gs-alpha) of eukaryotic adenylate cyclase (AC). ADP-ribosylation of this protein locks the Gs-a in an active conformation leading to persistent activation of the catalytic subunit of AC. The resultant accumulation of 3′5′-cyclic-AMP (cAMP) in cells exposed to CT has myriad effects on exposed enterocytes, but the critical event important to the diarrheal purge observed in cholera involves the activation of an apical ion channel called the cystic fibrosis transmembrane conductance regulator (CFTR) [15]. In brief, it is thought that protein kinase A (a cAMP activated kinase) phosphorylates CFTR and that this in turn activates the CFTR channel for ATP-dependent transport of chloride ion out of epithelial cells and into the intestinal lumen. This produces an osmotic imbalance that results in water movement out of the epithelium and into the intestinal lumen, producing watery diarrhea. Other *E. coli* enterotoxins (e.g., heat-stable toxin A or simply STa) which activate guanylate cyclase (GC) also cause acute diarrhea by activating CFTR through its phosphorylation [6]. CFTR may also be involved in other human diarrheal disease symptoms associated with bacterial and viral infection, neoplasm, or congenital inflammatory conditions. Transient blocking of CFTR may find applications in veterinary medicine as well because of serious outbreaks of post-weaning diarrhea in pigs where enterotoxigenic *E. coli* is the most common cause [16]. Thus, as a target for intervention, transient blocking of the CFTR channel function with pharmacological agents remains a promising option not only for cholera but for other diarrheal disease syndromes.

Previously, Verkman and colleagues reported the discovery of several small molecule compounds that showed promise as inhibitors of CFTR [17,18,19,20,21,22,23]. These molecules could block chloride secretion in cell culture and intestinal fluid accumulation in mouse models for diarrhea induced by intraluminal injection with pure toxins (either CT or STa). The first CFTR blockers had inhibitory potency in the 2–8 μM range, but these were designed to be impermeable and active on the apical side of polarized cells or the luminal side of intestinal epithelium [20]. One such molecule (iOWH032) was tested in Phase I studies but was not pursued because of its low potency and the fact that it was rapidly washed out of the intestine [17]. However, further medicinal chemistry efforts led to the development of more potent molecules such as the R enantiomer of benzopyrimido-pyrrolo-oxazinedione (BPO)-27. (R)-BPO-27 is active in the 4 nM range in cell culture assays and has been shown to inhibit channel function by competing for binding to the cytoplasmic side of the CFTR channel [21,22]. This mechanism of action was unique in that it stabilized the channel in a closed conformation with a cytoplasmic potency of ~600 pM. These promising results prompted studies to examine whether BPO-27 had the appropriate pharmacokinetics and potency to block enterotoxin-induced diarrheal disease. Indeed, BPO-27 showed greater than 90% oral bioavailability, no significant toxicity, and significant potency in blocking CT- and ST-induced fluid secretion in cultured human enteroids and mouse intestinal closed loops [23]. While these studies established the potency of BPO-27 when administered intraperitoneally or perorally, this work did not address whether BPO-27 could block secretory diarrhea in animals with an open intestine after exposure to a colonizing enterotoxic bacterial species. In this study, we investigated whether preoral administration of BPO-27 can block disease symptoms in a mouse model for acute diarrhea caused by toxigenic *V. cholerae*.

## 2. Results

### 2.1. BPO-27 Treatment Prevents Morbidity and Mortality Associated with Oral Administration of Cholera Toxin (CT) or Toxigenic Vibrio cholerae

Neonatal mice (3–5-day-old suckling pups) exhibit CT-dependent diarrheal disease and are considered the animal model “gold standard” for evaluating toxigenic *V. cholerae*-induced disease and other properties of infection such as bacterial intestinal colonization [24,25,26,27,28]. In this model, purified toxins or toxigenic bacterial species such as CT-producing *V. cholerae* are administered directly into the stomach via the oral–gastric route. Infected or CT-treated pups are then isolated in various treatment groups and assessed for disease and symptoms of morbidity. Animals were sacrificed at 18, 19, 20, or 22 h post-infection and diarrheal disease was determined by measuring the fluid accumulation ratio (FA). An elevated FA ratio is an indication of diarrhea disease in sucking mice [24,25] and is determined by dividing the weight of the entire gastrointestinal tract and its contents (i.e., the intact gut) by the remaining weight of the individual mouse after dissection and gut excision, as previously described [25,27]. This metric of diarrheal disease underestimates the actual diarrheal disease response to challenge with or without treatment in that it does not account for fluids lost from animals before gut extraction and measurement.

To test whether (R)-BPO-27 treatment reduces CT-dependent diarrheal disease symptoms, groups of 3–5-day-old wild type (CD-1) mice were mock treated with the vehicle control containing DMSO in Luria broth (LB), purified CT (List Biological Laboratories, Campbell, CA, USA), purified CT mixed with (R)-BPO-27, or with (R)-BPO-27 alone (Figure 1A). A second dose of vehicle or (R)-BPO-27 was administered 6 h after the initial challenge. Mice treated with CT alone had statistically significant elevated FA ratios (diarrheal disease) at 22 h (Figure 1B). By contrast, mice treated with CT mixed with (R)-BPO-27 developed significantly less diarrheal disease than mice treated with CT alone (Figure 1B). Importantly, there was no significant difference in diarrheal disease between mice treated with CT and (R)-BPO-27 and mice treated with (R)-BPO-27 alone or with the vehicle control. We conclude that two doses of (R)-BPO-27 protect neonatal mice from diarrheal disease induced by oral-gastric administration of CT. These results are consistent with the findings of Verkman and colleagues who studied the ability of (R)-BPO-27 to block fluid accumulation induced by CT and STa in isolated (i.e., ligated) intestinal segments of adult mice [23].

Next, we determined whether treatment with (R)-BPO-27 would lead to reduced diarrheal disease and morbidity in 4-day-old wild type CD-1 mice after oral–gastric infection with a CT-producing strain of *V. cholerae* (C6706). C6706 is a seventh pandemic El Tor isolate derived from a patient during the explosive and massive 1991 cholera epidemic in Lima, Peru [27]. To that end, 3–5 day-old suckling mice were challenged by the oral–gastric route with the vehicle control containing DMSO in Luria broth (LB), 1 × 10^8^ colony-forming units (CFU) of *V. cholerae* C6706, 1 × 10^8^ CFU of C6706 mixed with (R)-BPO-27, or 1 × 10^8^ CFU of C6706 mixed with (R/S)-BPO-27, an equal mixture of the R (active) and S (inactive) enantiomers of BPO-27 [22]. At 6 h post-infection, mice were treated again by the oral–gastric route with either vehicle, or (R)-BPO-27 or (R/S)-BPO-27 in the vehicle at the same initial dose levels. At 22 h post initial challenge, diarrheal disease was determined by measuring the FA ratio. Similar to what we observed with CT treatment, mice infected with CT-producing *V. cholerae* C6706 and treated with (R)-BPO-27 (or (R/S)-BPO-27) developed significantly less diarrhea than mice treated with *V. cholerae* C6706 alone (Figure 2A–C) indicating that both R enantiomer and the racemic form of BPO-27 were effective at blocking fluid accumulation induced by infection with *V. cholerae* C6706 in these neonatal animals.

Interestingly, both (R)-BPO-27 (or (R/S)-BPO-27) forms of BPO-27 showed a profound ability to protect C6706-challenged animals from morbidity (Figure 2D). In this first experiment, 100% of the animals treated with either form of BPO-27 survived, while all those treated with vehicle only were moribund or dead by 22 h. We decided to repeat this experiment and again observed 100% survival in groups of mice treated with either ®-BPO-27 or (R/S)-BPO-27, while 25% of the mice treated with vehicle alone were dead or moribund by 22 h. We also confirmed that the morbidity and mortality observed in this model depended on production of CT in that a nontoxigenic mutant of C6706 produced no observable morbidity and allowed 100% survival at 20 h post-challenge (data not shown). While some litter-to-litter to variation occurred during these experiments, we conclude that bo®(R)-BPO-27 and (R/S)-BPO-27 show a strong protective effect against morbidity and mortality in suckling mice challenged with toxigenic *V. cholerae* C6706.

### 2.2. BPO-27 Treatment Can Suppress Intestinal Colonization of V. cholerae

Besides causing the fulminant diarrhea characteristic of the disease cholera, CT has also been reported to enhance the intestinal colonization of *V. cholerae*, particularly when measured as CFU/g of colonic contents [29]. Colonic contents from infected mice are analogous to the ‘rice water stool’ that would be shed by cholera victims. Therefore, we wondered whether BPO-27 treatment could measurably affect the bacterial load in the colonic fluid of mice compared to vehicle treatment alone. The intestinal colonization factors of *V. cholerae* have largely been defined using the classical biotype strain O395 [24,26]. Accordingly, we measured the bacterial load in the colonic fluid of mouse pups challenged with *V. cholerae* O395. Treatment with both (R)-BPO-27 and (R/S)-BPO-27 had protective effects in that the compounds delayed the onset of diarrhea and the level of fluid accumulation measured at 18 h post-infection, although this trend did not reach clear statistical significance (Figure 3A–C). In previous work, fluid accumulation in this model with strain O395 has been shown to be solely due to production of CT [24,26], while colonization enhancement by CT produced by O395 in vivo has also been investigated in both rabbit [30] and human volunteer models [5]. Thus, it was interesting to observe that both the R and R/S compounds greatly reduced colonization of *V. cholerae* O395 in the colon in comparison to control vehicle treatment, with most animals showing no detectable CFU in the colonic fluid extracts (Figure 3D). The one outlier in the (R/S)-BPO-27 group that showed detectable colonization also showed a high level of fluid accumulation consistent with the concept that fluid secretion increases intestinal colonization [27]. However, there was no such correlation for the outlier in the (R)-BPO-27 group in that this mouse had a high bacterial load despite a low FA ratio. Although we observed some experimental variation, we conclude that two doses of both BPO-27 compounds significantly reduced the intestinal colonization of *V. cholerae* strain O395.

These encouraging results prompted us to develop a ‘treatment protocol’ where BPO-27 would only be administered after *V. cholerae* colonization of the small intestine was established. Previously, we had determined that by 5–6 h after challenge, *V. cholerae* CFU can be readily detected in the small intestine of 5-day-old infant mice [31]. Therefore, we tested whether BPO-27 treatment could reduce bacterial loads when given in a single dose 5 h post-challenge. To test this, we used *V. cholerae* C6706 as our challenge strain because BPO-27 compounds previously suppressed FA ratios induced by this strain when given in a two-dose regimen (Figure 1). When compared to vehicle control, we saw approximately a seven-fold reduction in bacterial intestinal colonization measured in the mouse small intestine (SI) in the (R)-BPO-27 treatment group. However, reduced colonization was not as observed in the racemic (R/S)-BPO-27 group or the vehicle control (Figure 4A,B). It is unclear whether R and S isomers of BPO-27 undergo some level of racemization in the intestine of 5-day-old mice, and thus lower potency of the (R/S)-BPO-27 was not entirely unexpected. We conclude that a single dose of the active (R) enantiomer of BPO-27 given early after infection can suppress bacterial colonization of the small intestine by *V. cholerae* C6706.

### 2.3. BPO-27 Treatment Promotes Host Survival Even after Challenge by the Virulent “Variant’ Strain of V. cholerae Haiti-1

Although the suppressions of fluid accumulation and bacterial colonization were interesting parameters to explore, we were most intrigued by the ability of (R)-BPO-27 and (R/S)-BPO-27 to reduce morbidity and mortality in the neonatal mouse model of cholera. Furthermore, given that the current seventh pandemic of cholera is dominated by ‘variant strains’ that have mutations in the CtxB gene that mimic CT alleles present in classical biotype strains, we wondered whether BPO-27 compounds could block mouse lethality due to these more contemporary isolates of *V. cholerae*. To address this question, we chose Haiti-1 (H1) as our challenge strain. This strain was isolated from a diarrheal disease victim early in the 2010 Haitian cholera epidemic [11]. By genomic sequence analysis, the H-1 strain is representative of the ‘variant’ *V. cholerae* strains that currently cause endemic and epidemic cholera worldwide [10,11]. For example, such strains have caused over one million cholera cases in both Yemen and Haiti with tens of thousands of deaths. Furthermore, the Haiti variant of *V. cholerae* has been reported to be hyper virulent in multiple animal models for cholera [32].

To explore whether oral administration of BPO-27 could be effective at preventing the morbidity or mortality after infection by *V. cholerae* H-1 that had already been established, we employed our treatment protocol that involved administration of a single dose of BPO-27 at 5 h post-challenge. As expected, *V. cholerae* H-1 indeed caused morbidity in almost half of the infected pups 20 h post-challenge. Remarkably, treatment with (R)-BPO-27 or (R/S)-BPO-27 at 5 h post-challenge fully eliminated morbidity compared to the vehicle control (Figure 5A,B). We conclude that both (R)-BPO-27 or (R/S)-BPO-27 are effective oral-gastric treatments for both preventing and treating the symptoms of cholera in this pre-clinical animal model that employs a hyper-virulent variant strain of *V. cholerae*.

## 3. Discussion

The compound (R)-BPO-27 has shown great promise as a potential therapy for diarrheal disease driven by activation of the CFTR channel [17,21,22,23]. Previous studies have established that this compound blocks CFTR channel function by locking it in the closed confirmation it adopts when not bound to ATP [21,22]. Thus, chloride secretion is inhibited even after the channel is activated by a cAMP- or cGMP-dependent phosphorylation. Verkman and colleagues first showed that BPO-27 could block the action of toxins such as CT and STa (which elevate cAMP and cGMP, respectively) in both cell culture models and in isolated intestinal segments (i.e., ligated intestinal loops) [23]. However, evidence that BPO-27 can block diarrhea driven by toxins released by a living enterotoxigenic bacterium colonizing the gut has not been previously reported.

In the studies described here, we obtained evidence for the efficacy of (R)-BPO-27 or (R/S)-BPO-27 in preventing fluid accumulation and death in a well-validated animal model for *V. cholerae* infection and disease. The 5-day-old neonatal mouse model mimics cholera in humans in that pups experience severe diarrheal disease symptoms a few hours after oral challenge by toxigenic *V. cholerae* and frequently become moribund within 18–24 h [24,26]. Remarkably, the protective effect of both (R)-BPO-27 and (R/S)-BPO-27 was not only observed when administered with the bacterial inoculum followed by a second dose hours later, but also by a single oral gastric dose of the compound administered 5 h post-challenge. We view the latter as a ‘treatment regimen’ since animals were likely exposed to CT expressed in vivo and are certainly already supporting rapid explosive growth of the toxigenic organism. Seeing efficacy in this treatment regimen bodes well for the use of BPO-27 in preventing morbidity and mortality from cholera under conditions where intravenous fluid replacement options are limited (e.g., explosive epidemics in the developing world). If the drug has few if any side effects in humans when administered orally, it could also be used as a preventative therapeutic in susceptible individuals that are exposed to the pathogen in the context of an epidemic. Although conventional antibiotics can also, in theory, be used as preventative therapeutics, their use can cause side effects through disruption of the intestinal microbiota as well as drive the development of problematic resistance in bystander bacteria. These negative properties of conventional antibiotics should not be a concern for BPO-27 given that as a drug it would target a host function rather than a bacterial property per se.

We also found that BPO-27 can suppress replication and survival of *V. cholerae* within the intestine, including the hyper-virulent Haiti-1 variant strain. Blocking the colonization-enhancing effects of CT is the most likely explanation for this observation [27]. Haiti-1 is similar to the most predominant strains of *V. cholerae* causing epidemic and pandemic cholera currently worldwide. While it is not known why variant strains have displaced older seventh pandemic El Tor biotype strains, these strains are generally more toxigenic and even have unique mutant alleles in the *ctxB* gene encoding the receptor-binding subunit of CT [32]. While such mutations may improve the fitness of *V. cholerae* as a human pathogen, our data suggest that BPO-27 treatment will nonetheless block the symptoms caused by this variant version of CT when delivered by a toxigenic variant strain.

An additional benefit to blocking the action of CT in the context of cholera is the likelihood that treated victims will shed few organisms. This is simply because the effects of CT on the intestinal mucosa cause changes that enhance the replication of *V. cholerae* [29]. In this report we provided evidence that BPO-27 indeed can reduce the bacterial load of *V. cholerae* in the colon and small intestine of infected animals, and this finding is consistent with the compound’s ability to block the action of cholera toxin at the level of CFTR activation. The ability of BPO-27 to suppress intestinal bacterial load might have a positive effect on controlling cholera epidemics because colonic bacteria burden likely translates into the bacterial load present in infectious cholera stools of patients.

## 4. Conclusions

In conclusion, our results support the advancement of BPO-27 to human Phase I safety studies and, if successful, to testing the compound in protective efficacy studies for *V. cholerae* oral challenge. The human subject volunteer challenge model is both safe and well established in the field and has been essential to understanding the efficacy and safety of live attenuated *V. cholerae* vaccines (PMID: 16028125). Unlike field trials, the human challenge model could quickly define the treatment window for BPO-27 administration and dosing that effectively prevents the severe diarrheal purge associated with cholera. Volunteer challenge models for enterotoxigenic *E. coli* (ETEC) also exist and would allow for assessment of the therapeutic potential of BPO-27 for blocking the activity of both heat-stable enterotoxins such as STa as well as *E. coli* heat-labile toxins that are highly similar to CT in their mode of action. However, although ETEC-mediated disease is much more common, particularly in travelers, “king cholera” remains the most feared threat to humans living on the edge of access to full modern hygiene and medicine and should be the ethical first choice for establishing the efficacy of BPO-27 as a therapeutic for severe diarrheal disease.

## 5. Materials and Methods

### 5.1. Bacterial Strains

The *V. cholerae* strains used in this study are listed in Appendix A. Unless indicated otherwise, bacteria were routinely grown aerobically at 37 °C in LB broth or on LB plates. Antibiotics were added to the medium at the following concentrations: 0.1 mg mL^−1^ streptomycin.

### 5.2. Drug Source and Properties

BPO-27 was provided by Vanda Pharmaceutics, Inc. in either its (R) or its (R/S) enantiomer forms as dry powders and each was greater than 98% pure. BPO-27 compounds were used immediately after dissolving in dimethyl sulfoxide (DMSO) and then diluted to the final concentrations as described below. (R/S)-BPO-27 is a racemic mixture of the BPO-27 compound, with the inactive S enantiomer comprising 50% of the total.

### 5.3. Suckling Mouse Experiments

All animal experiments in this study were approved by the Institutional Animal Care and Use Committee at Harvard Medical School (S00000116-6, Approved: 19 March 2021).

Suckling mice were infected with *V. cholerae* as previously described [24], with the following modifications. CD-1 mice were obtained from Charles River Laboratories. Three-to-five-day-old CD-1 sucking mice that had been separated from their mothers were randomly allocated to treatment groups and 2 h later were orally inoculated with 0.05 mL sterile LB broth (mock infection) or with 1 × 10^6^, 1 × 10^7^, or 1 × 10^8^ CFU of *V. cholerae* in 0.05 mL LB broth or with 10 μg purified cholera toxin in 0.05 mL LB broth. When co-gavaging with BPO-27 or vehicle, this inoculum media also included 2.74% DMSO with 2.5 μg/g (R)-BPO-27 or 5 μg/g (R/S)-BPO-27.

A second oral gavage was performed either 5 or 6 h post-infection, where mice were given 0.05 mL sterile LB broth with 2.74% DMSO (vehicle), 0.05 mL sterile LB broth with 2.74% DMSO and 2.5 μg/g (R)-BPO-27, or 0.05 mL sterile LB broth with 2.74% DMSO and 5 μg/g (R/S)-BPO-27.

At the indicated time point, suckling mice were euthanized, and gastrointestinal tracts were collected for bacteriological analysis and to determine diarrheal disease. The fluid accumulation ratio in mice was determined by the weight of the entire gastrointestinal tract (gut)/(mouse body weight − gut weight), as previously described [29]. Group sizes (at least four mice per treatment group) represent the minimum number required to reach statistical significance (*p* < 0.05) between experimental groups. For diarrheal onset determination, mice were monitored every 30 min for the entire duration of the experiment. Diarrheal onset after oral treatment with *V. cholerae* or treatment with CT was determined by the presence of diarrheal fluid spots on the bottom of the mouse container.

Mice were determined to be moribund as per IACUC guidelines. Mice that had impaired mobility, persistent recumbency (inability to rise), labored breathing, lack of response to external stimulus, loss of color, neurological signs, or severe weight loss (>20%) were determined to be moribund and euthanized. All euthanized animals were included in the survival analysis.

### 5.4. Statistical Analysis

No statistical methods were used to predetermine sample size. Statistical analyses were performed with the Prism software (GraphPad). An unpaired two-tailed Student’s *t*-test was used to determine whether differences between two groups were statistically significant (*p* < 0.05). For comparisons between three or more groups, a one-way ANOVA (between-subjects factors) or two-way ANOVA (between-subjects factors) was used followed by post hoc multiple comparisons test.

## Figures and Tables

**Figure 1 toxins-14-00225-f001:**
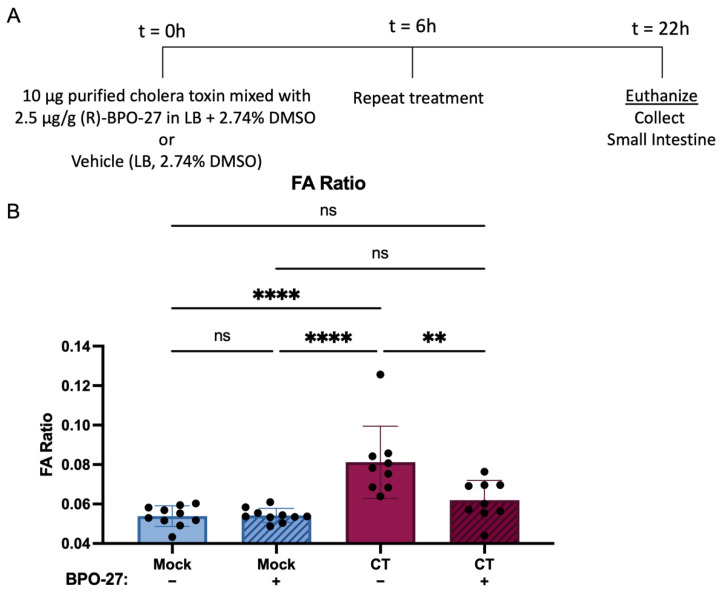
BPO-27 reduces CT-dependent diarrheal disease symptoms in the suckling mouse model. (**A**) Groups of mice were mock treated, or treated orally with purified CT, purified CT mixed with (R)-BPO-27, or with (R)-BPO-27 alone. Six hours after treatment, mice were mock treated or re-administered (R)-BPO-27. Then, 22 h after treatment, the gastrointestinal tract was weighed, and the fluid accumulation ratio was determined. (**B**) The fluid accumulation ratio of the groups described in (**A**) was calculated as the weight of the gut divided by the remaining body weight. Fluid accumulation ratios were compared using a one-way analysis of variance (ANOVA) (F = 13.51, *p* < 0.0001) followed by Tukey’s multiple comparisons test. Column heights represent group means and error bars represent standard deviation. Each dot represents an individual mouse. ** *p* ≤ 0.01; **** *p* ≤ 0.0001; ns, not statistically significantly different (*p* > 0.05).

**Figure 2 toxins-14-00225-f002:**
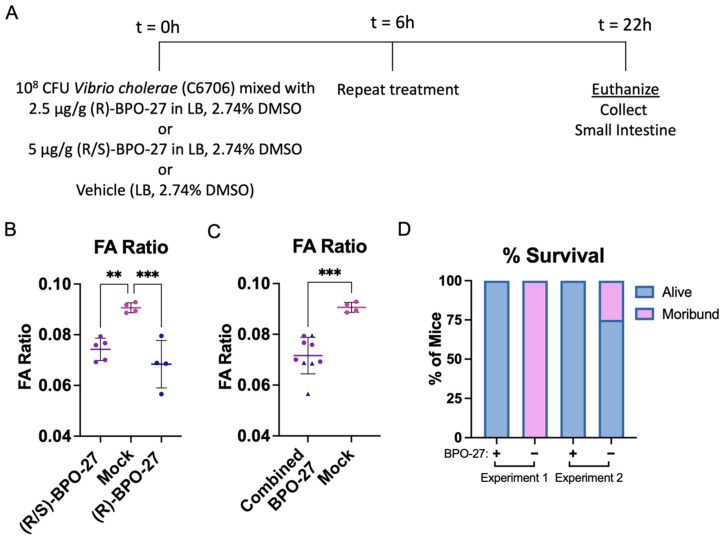
BPO-27 blocks diarrheal disease and morbidity from toxigenic *V. cholerae* in the suckling mouse model. (**A**) Groups of mice were mock treated, infected orally with 10^8^ CFU *Vibrio cholerae* C6706, infected orally with 10^8^ CFU of *Vibrio cholerae* C6706 mixed with (R)-BPO-27, infected orally with 10^8^ CFU of *Vibrio cholerae* C6706 mixed with (R/S)-BPO-27, treated orally with (R)-BPO-27 alone, or treated orally with (R/S)-BPO-27 alone. Six hours post initial treatment, mice were mock treated again or re-administered (R)-BPO-27 or (R/S)-BPO-27. (**B**) The fluid accumulation ratio of the treatment groups described in (**A**). Fluid accumulation ratios were compared using a one-way analysis of variance (ANOVA) (F = 15.32, *p* = 0.0009) followed by Tukey’s multiple comparisons test. Lines represent group means and error bars represent standard deviation. (**C**) The combined fluid accumulation ratio of the (R)-BPO-27 and (R/S)-BPO-27 treatment groups. Fluid accumulation ratios were compared using an unpaired *t*-test (*p* = 0.0004). Lines represent group means and error bars represent standard deviation. (**D**) The survival of the mice from groups described in (**A**). No mouse treated with BPO-27 was considered moribund at the endpoint, while 66.6% of vehicle-treated mice were scored as moribund. Each dot and triangle represents an individual mouse. ** *p* ≤ 0.01; *** *p* ≤ 0.001; ns, not statistically significantly different (*p* > 0.05).

**Figure 3 toxins-14-00225-f003:**
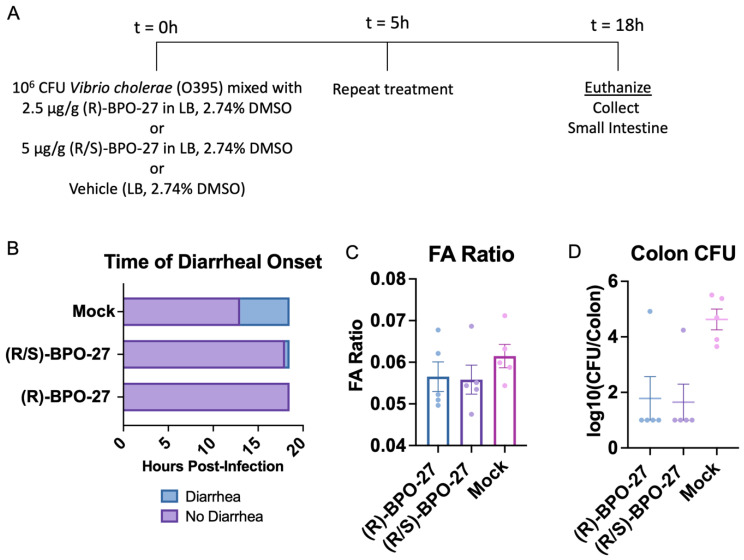
BPO-27 compounds reduce intestinal colonization of *V. cholerae* O395. (**A**) Groups of mice were mock treated, infected orally with 10^6^ CFU of *Vibrio cholerae* O395, infected orally with 10^6^ CFU of *Vibrio cholerae* O395 mixed with (R)-BPO-27, or infected orally with 10^6^ CFU of *Vibrio cholerae* O395 mixed with (R/S)-BPO-27. Five hours post initial treatment, mice were mock treated again or re-administered (R)-BPO-27 or (R/S)-BPO-27. (**B**) Time of diarrhea onset from mice in groups described in (**A**). (**C**) The fluid accumulation ratio from mice in groups described in (**A**). Groups were compared using a one-way analysis of variance (ANOVA) (F = 0.8745, *p* = 0.4420) followed by Tukey’s multiple comparisons test. Lines represent group means and error bars represent s.e.m. (**D**) The bacterial CFU (colony-forming units) per colon were determined from the colon contents of mice. All mock-treated mice had detectable bacterial colonization, while 80% of the BPO-27-treated mice had colonization below the limit of detection. Each dot represents an individual mouse.

**Figure 4 toxins-14-00225-f004:**
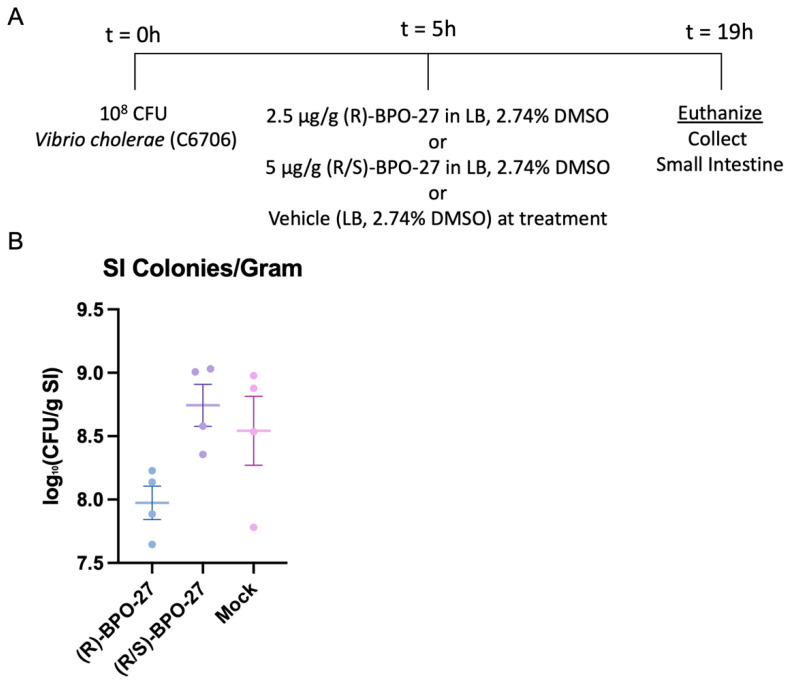
Treatment with BPO-27 reduces intestinal colonization of toxigenic *V. cholerae*. (**A**) Groups of mice were mock infected or infected orally with 10^8^ CFU of *Vibrio cholerae* strain C6706 and five hours post-infection, mice were mock treated, treated with (R)-BPO-27, or treated with (R/S)-BPO-27. (**B**) The bacterial CFU (colony-forming units) per colon was determined from the colon contents of mice 19 h post-infection. The fluid accumulation ratio of the three different treatment groups were compared using a one-way analysis of variance (ANOVA) (F = 4.041, *p* = 0.0559) followed by Tukey’s multiple comparisons test. Lines represent group means and error bars represent s.e.m. Each dot represents an individual mouse.

**Figure 5 toxins-14-00225-f005:**
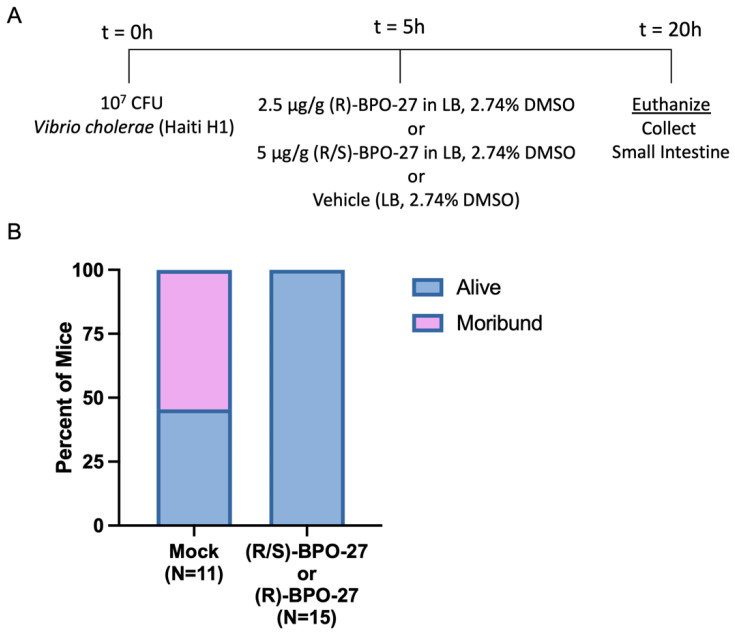
Treatment with BPO-27 increases survival of mice after challenge with *V. cholerae* Haiti-1. (**A**) Groups of mice were mock infected or infected orally with 10^7^ CFU of *Vibrio cholerae* Haiti H1 and five hours post-infection, mice were mock treated, treated with (R)-BPO-27, or treated with (R/S)-BPO-27. (**B**) The survival of mice was assessed at 20 h post-infection, with morbidity assessed as previously defined. No mice treated with BPO-27 were considered moribund at the experimental endpoint, while 54.5% of the mock-treated mice were determined to be moribund.

## Data Availability

The datasets in this study are available from the corresponding author upon reasonable request.

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
