# Peer review of "A Potent Inhibitor of the Cystic Fibrosis Transmembrane Conductance Regulator Blocks Disease and Morbidity Due to Toxigenic *Vibrio cholerae"

_toxins, 2022, doi:10.3390/toxins14030225_

Round 1

Reviewer 1 Report

TO AUTHORS

This article deals with important issues about how it is possible to combat lethal bacterial diarrheal disease caused by the effects of cholera toxin and related enterotoxins. The authors explain excellently how blocking the function of the CFTR channel with pharmacological agents may be a promising option in blocking the effects of cholera and other diarrheal disease syndromes. Using the BPO-27 compound previously described by Verkman and colleagues, the authors have demonstrated the good potential of using the BPO-27 compound in suppressing the replication and survival of V. cholera within the intestine. The method used excellently demonstrates the effectiveness of using the BPO-27 compound in preventing fluid accumulation and death in a well-validated animal model for V. cholera infection, proposing the development of this therapy as a good treatment protocol. In conclusion, I think that the paper can be considered for publication without revision.

Author Response

RESPONSE: We want to thank the reviewer for the positive review of our manuscript.

Reviewer 2 Report

The authors present a prospective investigation that assesses the efficacy of a new therapy to block cystic fibrosis transmembrane conductance regulator (CFTR) as a method to attenuate the effects of cholera toxin (CT) released from Vibrio cholerae. The investigation uses a murine model of diarrhea. The compound of interest was BPO-27, which was administered in two forms: (R)-BPO-27 and (R/S)-BPO-27. In addition to improving or blocking diarrhea, the compounds improve mortality. Lastly, the compound prevents replication of the bacterium in the gut.

More than one strain of V. cholerae were used, including one particularly lethal strain.

In general, the work is well-written and the experiments designed appropriately. The number of animals utilized likely provided adequate statistical power given the differences in mean and SD. The statistical methods were also sound. Given the software, I would like the authors to indicate what the power was post hoc, which should be provided in the output of the program used.

Indicate in the Abstract the full names of (R)-BPO-27 and (R/S)-BPO-27.

I noticed that there was a spectrum of wellness that was not defined in the Methods. Some mice are alive. Some moribund. Some dead. There are no indices of activity, vital signs, color, etc. When does a moribund mouse become dead? Onset of diarrhea also presents a problem. Was there someone there for essentially 24 hours checking if the mouse was dead or having diarrhea? The entire in vivo survival/diarrhea series is seriously lacking in objective endpoints.

Figure 2b and 2c have the labels of the x-axis cut off. 2b-c are also small panels that are hard to read.

Figure 5b is has the labels of the x-axis partially cut off.

Line 400. “Mice that were euthanized before the end of the experiment because of health concerns were excluded from the analysis.” How many mice were excluded per group because of this reason? How many groups with no morbidity or mortality had mice euthanized this way? Did any of the treatment mice die before being counted?

In sum, while promising, there seems to be a great deal of subjective or poorly documented endpoints concerning the status of animals after infection. Also, the idea that those that did not make it to the end of the experiment secondary to “health concerns” is perhaps the point of the experiment. More information is needed to adequately assess this work.

Author Response

Responses are shown in BLUE.

Reviewer 2:

The authors present a prospective investigation that assesses the efficacy of a new therapy to block cystic fibrosis transmembrane conductance regulator (CFTR) as a method to attenuate the effects of cholera toxin (CT) released from Vibrio cholerae. The investigation uses a murine model of diarrhea. The compound of interest was BPO-27, which was administered in two forms: (R)-BPO-27 and (R/S)-BPO-27. In addition to improving or blocking diarrhea, the compounds improve mortality. Lastly, the compound prevents replication of the bacterium in the gut.

More than one strain of V. cholerae were used, including one particularly lethal strain.

In general, the work is well-written and the experiments designed appropriately. The number of animals utilized likely provided adequate statistical power given the differences in mean and SD. The statistical methods were also sound. Given the software, I would like the authors to indicate what the power was post hoc, which should be provided in the output of the program used.

RESPONSE: We thank the reviewer for their positive and helpful comments which have helped us to improve our manuscript.

Indicate in the Abstract the full names of (R)-BPO-27 and (R/S)-BPO-27.

RESPONSE: Thank you for this suggestion. We have added the full names of (R)-BPO-27 and (R/S)-BPO-27 in the abstract.

I noticed that there was a spectrum of wellness that was not defined in the Methods. Some mice are alive. Some moribund. Some dead. There are no indices of activity, vital signs, color, etc. When does a moribund mouse become dead? Onset of diarrhea also presents a problem. Was there someone there for essentially 24 hours checking if the mouse was dead or having diarrhea? The entire in vivo survival/diarrhea series is seriously lacking in objective endpoints.

RESPONSE: Thank you for pointing this out. Humane endpoints used are based on criteria listed in our Institutional Animal Care and Use Committee (IACUC) proposal which were used to determine the end of experimental studies so as to avoid or terminate unrelieved pain and /or distress experienced by a moribund animal. We have experience with determining these humane end-points indicating an animal is moribund. Specifically, animals that have impaired mobility, persistent recumbency (inability to rise), labored breathing, lack of response to external stimulus, loss of color, neurological signs, or severe weight loss (>20%) are deemed to be moribund. We have added more information on how mice were determined to be moribund to the Methods section.

Yes – for the “onset of diarrhea” experiment, mice were checked every 30 minutes for the entire duration of the experiment. Diarrheal onset was determined based on our experience with this model after oral treatment with V. cholerae or treatment with CT and can be clearly observed by the presence of liquid yellow-brown diarrheal spots on the bottom of the container, which is never present in containers of uninfected or healthy animals. We have added more information on how diarrhea is assessed in the Methods section. We thank the reviewer again for their helpful comments.

Figure 2b and 2c have the labels of the x-axis cut off. 2b-c are also small panels that are hard to read.

RESPONSE: Thank you for pointing this out. We are able to see the axes in all of our figures, so we are not sure if this was an error that occurred during / after submission. We have modified the figures so hopefully this issue is resolved.

Figure 5b is has the labels of the x-axis partially cut off.

RESPONSE: See response above.

Line 400. “Mice that were euthanized before the end of the experiment because of health concerns were excluded from the analysis.” How many mice were excluded per group because of this reason? How many groups with no morbidity or mortality had mice euthanized this way? Did any of the treatment mice die before being counted?

RESPONSE: Thank you for pointing out this mistake in the language used in our methods, which is incorrect based on the design of this study. The sentence “Mice that were euthanized before the end of the experiment because of health concerns were excluded from the analysis” is one we typically use for experiments using animals since our IACUC protocol does not allow death as an end-point. However, in this study, mice that were euthanized because of health concerns (i.e moribund) were obviously not excluded from the analysis since we used this to determine the percent survival. Only mice that died immediately after oral gavage or from isoflurane exposure were excluded from analysis in this study. It’s important to emphasize that no mice that survived the initial oral gavage were excluded from the morbidity analysis in our experiments and all BPO-27-treated mice were included in the survival analysis.

In sum, while promising, there seems to be a great deal of subjective or poorly documented endpoints concerning the status of animals after infection. Also, the idea that those that did not make it to the end of the experiment secondary to “health concerns” is perhaps the point of the experiment. More information is needed to adequately assess this work.

RESPONSE: We believe that our manuscript now contains adequate detail in the methods to address these concerns. We thank the reviewer again for their helpful comments which have helped us to improve our manuscript.

Reviewer 3 Report

  1. Fig2, Fig5 captions are shifted and unreadable
  2. There is no explanation in the text that (R/S)-BPO-27 is a mixture of R and S forms of BPO-27 enantiomers
  3. Line 177-178. «...(Fig. 2A, Fig. 2B and Fig. 2C), indicating that both the R-enantiomer and racemic form BPO-27 effectively block fluid accumulation...»

2.5 mg/g R-BRO-27 and 5 mg/g (R/S)-BPO-27 were used. Is 5 mg/g (R/S)-BPO-27 contains 2.5 mg/g R-BRO-27? Could the effect of 5 mg/g (R/S)-BPO-27 be an effect of included 2.5 mg/g R-BRO-27? In the case the S-BRO-27 compound is inactive, isn’t it?

Author Response

Responses are shown in BLUE.

Reviewer 3:

We thank the reviewer for their positive and helpful comments which have helped us to improve our manuscript.

  1. Fig2, Fig5 captions are shifted and unreadable

RESPONSE: Thank you for pointing this out. We are able to see the captions in all of our figures, so we are not sure if this was an error that occurred in the system during / after submission. We have modified the figures so hopefully this issue is resolved.

  1. There is no explanation in the text that (R/S)-BPO-27 is a mixture of R and S forms of BPO-27 enantiomers

RESPONSE: Thank you for pointing this out. We have added this to the text and methods.

  1. Line 177-178. «...(Fig. 2A, Fig. 2B and Fig. 2C), indicating that both the R-enantiomer and racemic form BPO-27 effectively block fluid accumulation...»

2.5 mg/g R-BRO-27 and 5 mg/g (R/S)-BPO-27 were used. Is 5 mg/g (R/S)-BPO-27 contains 2.5 mg/g R-BRO-27? Could the effect of 5 mg/g (R/S)-BPO-27 be an effect of included 2.5 mg/g R-BRO-27? In the case the S-BRO-27 compound is inactive, isn’t it

RESPONSE: Yes, that is correct. Previous work has shown that the S enantiomer is inactive. The racemic (R/S)-BPO-27 mixture contains 50% of each enantiomer, so 5 mg/g (R/S)-BPO-27 is equal to delivering 2.5 mg/g of the bioactive R-BPO-27. We hope that future work can utilize the R/S mixture and thus avoid the enantiomer purification step in the synthesis of the compound.

We thank the reviewer for pointing out that this needed clarification. We have added this additional information to the main text and methods.

Round 2

Reviewer 2 Report

No further comments.